Today’s computing challenges: opportunities for computer hardware design

http://orcid.org/0000-0002-9274-0182 Bae Woorham 1 2 wrbae@eecs.berkeley.edu
1 Circuits Department, Ayar Labs , Santa Clara, CA , USA
2 Department of Electrical Engineering and Computer Sciences, University of California, Berkeley , Berkeley, CA , USA
Somani Arun
Electronic publication date: 2021 Mar 30
Publication date: 2021
Volume: 7
Electronic Location ID: e420
Received 2020 Nov 4; Accepted 2021 Feb 9
Copyright: © 2021 Bae
Copyright year: 2021
Copyright holder: Bae
License: This is an open access article distributed under the terms of the Creative Commons Attribution License, which permits unrestricted use, distribution, reproduction and adaptation in any medium and for any purpose provided that it is properly attributed. For attribution, the original author(s), title, publication source (PeerJ Computer Science) and either DOI or URL of the article must be cited.
License URL: https://creativecommons.org/licenses/by/4.0/

Keywords: Computer, Hardware, Silicon, Processor, Memory, Interconnect, Moore’s law, Circuit design

Funding: The authors received no funding for this work.

==============================
Due to the explosive increase of digital data creation, demand on advancement of computing capability is ever increasing. However, the legacy approaches that we have used for continuous improvement of three elements of computer (process, memory, and interconnect) have started facing their limits, and therefore are not as effective as they used to be and are also expected to reach the end in the near future. Evidently, it is a large challenge for computer hardware industry. However, at the same time it also provides great opportunities for the hardware design industry to develop novel technologies and to take leadership away from incumbents. This paper reviews the technical challenges that today’s computing systems are facing and introduces potential directions for continuous advancement of computing capability, and discusses where computer hardware designers find good opportunities to contribute.

Introduction

These days, the world has been evolving very fast in various areas. The focus of technology development has been also switching to realize better human experience, convenience, and happiness, rather than old focuses such as mass production, automation, or cost reduction. Such rapid changes severely impact on the silicon industry, which has been responsible for the computing capability of the planet for several decades. The impact can be either positive or negative; it can provide more opportunities but also introduce many challenges at the same time. In fact, the opportunities are all about data (Horowitz, 2014; Kim, 2015). That is mainly because the world needs more electronics to handle the data. As mentioned above, the world is pushing to realize a whole bunch of things (such as smart cities, security, autonomous vehicles…) for better human experience, convenience, and happiness. In order to do that, we need to create, replicate, and process all the data. Accordingly, all the surveys predict that the amount of digital data will increase exponentially in the next 10 years. For example, Fig. 1 shows Cisco’s two reports on the amount of data creation in the world, which were released in 2017 and 2019 (Cisco Visual Networking Index, 2017; Cisco Visual Networking Index, 2019). Both reports predict that the amount of data will grow exponentially, but the 2019 report tells that the data has been created more than that expected 2 years before, and it will increase more rapidly.

Figure 1 Amount of digital data creation trend and projection (source: Cisco (Cisco Visual Networking Index, 2017, 2019)).

The world is driven by data, and electronics are responsible for handling those data, which means that we need to create more and more electronics devices. Cisco predicts that the number of electronic devices will increase by almost twice in 5 years. Nvidia gives a bit more aggressive prediction, such that the number of total connected devices will increase by 16 times in 7 years (Shao, 2019). No matter how much it is, everybody expects that there will be more needs for electronic devices.

On the other hand, the challenges are also all about data. Here are some critical concerns one can raise in the age of such exploding data. How do we process such amount of data? Where should we store the data? How do we communicate with the data? And, what happens if we keep the same energy efficiency while the amount of data is exploding? Going back to the Fig. 1, where the Cisco’s projection is shown, the amount of digital data is going to explode. If so, what happens if we keep the same energy efficiency for processing, storing, and communicating? Then the energy consumption will increase at the same rate as the data explosion, which is definitely not affordable. It is known that we are already consuming the largest portion of energy in the Earth for handling the data with electronics (Bae, Jeong & Jeong, 2016; Whitney & Delforge, 2014; Pierce, 2018); definitely such amount of data should not be affordable. From this observation, we would say that the energy efficiency must be improved proportional to the data explosion, at least to keep the same amount of energy consumption.

In fact, such explosion of data is not something that started yesterday, even though there might be some difference in degree. Hence, it is worthwhile trying to learn something from the history, how we did handle such exploding data before. Figure 2 shows a simplified computing system, where we can see a logic (processor) IC and a memory IC, and an interconnect link between them. Basically, in order to handle more data, we need higher processing speed, interconnect bandwidth, and memory density. Figure 2 also shows a simple summary of how we managed to enable it. For the processor side, the CMOS (complementary metal-oxide silicon) technology scaling, which is generally represented by Moore’s law (Moore, 1965, 1975), enables a transistor to be faster and consume even less power (Holt, 2016; Bohr & Young, 2017; Mak & Martins, 2010; Yeric, 2019). Once we have a faster transistor, we can raise the clock rate for faster processing. Once after the power scaling of transistor has been retarded due to some physical reasons (i.e., leakage current), people introduced parallelism such as multi-core processing to increase the processing speed without increasing the clock rate (Danowitz et al., 2012). For the memory side, the scaling of device footprint enabled a higher memory density (Hwang, 2002). However, extensive scaling led to many challenges, which were overcome by the memory industry with process innovations such as higher aspect ratio of DRAM, and material innovations like high-k materials (Mueller et al., 2005; Sung et al., 2015; Jang et al., 2019). For the interconnect side, the transistor scaling has been also a key enabler for a higher bandwidth, because a faster transistor makes a circuit faster (Daly, Fujino & Smith, 2018; Horowitz, Yang & Sidiropoulos, 1998). However, the electrical channels (wires) which bridge separate ICs cannot be scaled with the silicon technology, as they present in the physical world, not in the digital IC world. That is, an electrical channel has a finite bandwidth so that high-frequency components of transmitted signal attenuate over the channel. As a result, interconnect engineers had to make many innovations in equalization circuits which compensate the channel loss at high frequency, that is to equalize the channel response at low and high frequency (Horowitz, Yang & Sidiropoulos, 1998; Dally & Poulton, 1997; Zerbe et al., 2003; Stojanovic et al., 2005; Choi, Hwang & Jeong, 2004). They also introduced time-interleaving technique, which is something like the parallelism, to achieve very high speed even above the transistor limit (Kim & Horowitz, 2002; Lee, Dally & Chiang, 2000; Musah et al., 2014; Bae et al., 2017).

Figure 2 Summary of how we have made a better computer and why that will not work for future computers.

However, these legacy approaches cannot be good solutions for these days and the future. First of all, we are about to lose the almighty scaling. The scaling has not been fully finished yet; however, it has been a while since the power scaling started being retarded as discussed earlier. As a result, increasing the clock frequency is no longer available because we do not want to burn the chip out. The parallelism was introduced to overcome such challenge, but it has also hit the limit because of the same heat dissipation issue. Only a fraction of the multi-cores can be turned on at the same time, which is called “dark silicon” (Yeric, 2019; Esmaeilzadeh et al., 2011).

The similar issue happened to the memory, that is, the scaling has been retarded which limits the increase in the memory density. The scaling also introduced many non-idealities so that there are many higher-level assistances which burden memory module and increase the latency of the memory. For the interconnect side, the channel loss becomes very significant as the required interconnect bandwidth increases, so the equalization circuitry consumes too much power. It will be tougher as the scaling is ending because we can no longer take advantage of faster transistors. To summarize this section, the legacy solutions for handling data explosion will not be as effective as they used to be for the today’s and future computer. From the following section, we will discuss on the possible solutions that enable the continuous advance in computing capability for the next 10 years.

The remainder of this article is organized as follows. “Logic (System Semiconductor)” presents potential solutions for addressing the challenges of the logic and opportunities for the computer hardware design industry and engineers. “Memory and Storage” describes the recent innovations from the memory industry and discusses the future direction. In addition, the opportunities for design engineers from the revolution of the memory devices will be discussed. “Interconnect” discusses the recent trend of the interconnect technology and potential solutions to resolve the challenges that the interconnect technology is facing. Finally, conclusions are provided in “Conclusions”.

Survey methodology

This review was conducted in Sept.–Oct. 2020. Three different approaches were used to collect research articles:Searching Google scholar and IEEE Xplore with various keywords such as Moore’s law, CMOS scaling, high-bandwidth memory, V-NAND, crosspoint memory, transceiver, PAM-4, and silicon photonics.

Starting from an initial pool of articles and then move back and forth between their citations and references.

Selecting articles based on their impact and credibility; Prioritizing articles with high citations or from top conferences and journals of the fields, such as JSSC, TCAS, TCAD, TED, AELM, ISSCC, VLSI, IEDM.

Logic (system semiconductor)

Efficient computing with specialized IC

In this section, the technology directions for silicon logic to maximize the opportunity for the hardware design is discussed. While dealing with technology development of computation logic, it is inevitable to discuss the scaling limit of semiconductor process technology, the end of Moore’s law. In fact, since 2014, there have been at least one of plenary talks at International Solid-State Circuits Conference (ISSCC) that discuss on preparing the end of the Moore’s law, for example Kim (2015) and Vandersypen & van Leeuwenhoek (2017). So, let us take a quick look at where the scaling limit comes from. In his talk at ISSCC2015 (Kim, 2015), the president of Samsung Electronics told that the physical limit of transistor dimension is around 1.5 nm, which is given from Heisenberg’s uncertainty principle. However, he also told that he expected that the practical limit will be 3 nm. After 5 years, now, the 7 nm technology is already widely available in the industry. And the leading foundries such as TSMC and Samsung Electronics are already working on 5 nm and 3 nm technology development, which means that we are almost there.

As a result, recalling the energy discussion in the “Introduction”, the appropriate question for this point should be how we can improve the energy efficiency without scaling. We can find some hints from today’s mining industry, the cryptocurrency mining, where the computing energy efficiency is directly translated to the money. Recalling 2017, when the cryptocurrency value hit the first peak, the readers may remember that the graphics processing unit (GPU) price became very expensive. It is simply because the GPU is much more efficient than central processing units (CPU), so mining with GPU gave more profit margin. Then, why the GPU is much efficient compared to CPU?

It is because it is specialized. CPU is more generic, but the GPU is more specific. That is, there is a computing trade-off between the flexibility and the efficiency. After finding that, people went to field-programmable gate array (FPGA) for cryptocurrency mining for better efficiency, and eventually they end up with designing application-specific integrated circuit (ASIC) just for the mining. Figure 3 shows the survey of various cryptocurrency miners, where we can find an ASIC miner provides 104 times better efficiency than a CPU miner. From the observation, we can conclude that such a huge gain comes from the design of specialized ICs. To summarize, making specialized ICs is one of the top promising solution for the efficient computing. In accordance with that, the foundry companies would diversify their process technology instead of scaling it down, for example the Global Foundries 45 nm CLO process, which is specialized to silicon photonics (Rakowski et al., 2020).

Figure 3 Survey of energy efficiency of various cryptocurrency miners.

Productivity problem of specialization

We found that the specialization would be a potential solution to resolve the energy problem and to retain the continuous advance of computing. However, there are also some downsides of the specialization, so we need to investigate how profit is made in the new age with the specialization. In a simplified model in Fig. 4A, a fabless company shipped 1 million units of a generic chip before, but they are planning to design 10 specialized chips in 10 different processes to meet the better efficiency requirement. At the same time, they are expecting they can ship 2 million chips in total as there will be more demand of electronics. In the model, the company is currently making $3 million profit. On the right side of Fig. 4A, a linear extrapolation is made to when the company designs 10 specialized chips and total shipping is doubled. Note that all the cost is extrapolated in linearly proportional to the amount of production.

Figure 4 Productivity issue of specialization.

Case study of (A) ideal case, (B) practical case, (C) practical case with reduced design time.

However, it is too optimistic projection. Figure 4B shows a bit more realistic model. The revenue and production cost are indeed proportional to the amount of shipping. However, does it make sense to extrapolate other expenses? Of course, the answer is no. For example, the amount of manpower cannot be scaled linearly. To design a single complete chip, they need analog engineers, digital engineers, manufacturing engineers and more. Therefore, it makes no sense that only 4 engineers can make a chip which used to be made by 20 engineers, even though there must be some amount of efforts that can be shared among the chip designs. So, the model in Fig. 4B assumes 10 engineers can design a specialized chip A0. If so, the profit becomes minus. The calculation here is very rough, but at least we can observe a large fraction of design cost is not scaled with the amount of production. The company would raise the price, but customers will not be happy with that. Then, is the specialization a false dream?

The most reasonable solution here is to reduce the design time, since such design costs are proportional to the design time, as shown in Fig. 4C. For example, if they can reduce the design time by half, they can reduce the expenses by half, then they can make more profit. As mentioned earlier, they are designing 10 different but similar chips, and there is some amount of sharable efforts. That means, if they maximize the amount, they should be able to reduce the design time considerably.

Reducing design time by reusing design

Then what should we try to maximize shareability? Generally speaking, we can say the analog and mixed signal (AMS) circuit design is usually the bottleneck of reducing design time. That is mainly because AMS circuit designs highly rely on human’s heuristic knowledge and skills, compared to the digital design. Moreover, the design complexity has been increased as technology scales down, due to the complex design rules and digital-friendly scaling of CMOS technology, which is represented by the number of design rules shown in Fig. 5A, where we can find the design complexity has been increased exponentially as the technology scales down (White; Whitcombe & Nikolic, 2019; Han et al., 2021). Figure 5B shows a general design flow of an AMS circuit. Once we decide a circuit topology, we carefully size the transistor dimensions based on some calculations, and run simulation using CAD tools. If the simulation result is not positive, we go back and tweak the sizing. Once we meet the spec with the schematic simulation, we proceed to draw the layout mask, after that we run parasitic extracted (PEX) simulation and check the result again. Based on the result, we have to go back and forward many times until the performance of the circuit is fully optimized. The main issue here is that most of time is spent for drawing layout, and its complexity has been increasing as shown in Fig. 5A. One may ask why we do not try to automate the analog design as we did for the digital design. However, in fact, it is hard to say we can do it for the layout design in the near future because there are only a few ways to do it right, however there are billions of ways to do it wrong. That means, to make the automation tool work correctly, a designer should constrain the tools very precisely (Habal & Graeb, 2011; Lin, Chang & Lin, 2009), so they spend most of the design time constraining the tools, which is not very efficient (Chang et al., 2018). That is the main reason why the engineers in this field rarely use such automation tools.

Figure 5 (A) Silicon design complexity across technology node, (B) general design flow of AMS circuit.

In fact, a better way is reusing, because reuse is a bit easier than automation. For example, we can just grab a good designer who knows how to do it correctly and let him/her do the design. At the same time, we enforce him/her to write down every single step he would do to create a correct output into an executable script (often called as a generator). Then the script has the way of doing right of the good designer, so the output should be correct no matter who run the script. However, because transistor shapes are different between process technologies, it is hard to automatically capture a design-rule-compatible shape only with the script, without intervention of designer’s heuristic knowledge. Therefore, such script-based approach works well in a single process technology, however it would face many challenges when ported to another technology. To address such portability issue, template-based approaches have been proposed in many works (Crossley et al., 2013; Yilmaz & Dundar, 2008; Castro-Lopez et al., 2008; Martins, Lourenco & Horta, 2013; Kunal et al., 2019; Wang et al., 2019). Instead of letting a layout script draw a layout from scratch, designers prepare design-rule-aware templates of primitive components. The script assembles the templates by following the way an expert designer pre-defined. It is like a Lego block, when we buy a Lego package, there are many unit blocks (templates) and an assembly manual (script), as shown in Fig. 6.

Figure 6 Design flow based on reuse of design process using executable script (generator): (A) for the first process, (B) for porting designs to another process.

Such reuse-based approach is very attractive for the future of specialization, however there are some hurdles that the designers must overcome. In fact, the hurdle is not a matter of development of elegant CAD tools. Here is an example based on the author’s engineering experiences. The author has used three different frameworks for helping such reusing process, the Laygo, XBase, and ACG (Berkeley Analog Generator, 2021a, 2021b; Ayar Custom Generator, 2021). They are quite different each other as summarized in Fig. 7, for example the Laygo defines the templates more strictly so it more limits the degree of freedom, whereas the ACG has loose template definitions. There are pros and cons; the Laygo reduces the number of ways to do in wrong way for easier portability at the cost of sacrificing the degree of freedom. The ACG allows freedom however it burdens a designer spend more time on writing a portable script. That is, to summarize, there is just a trade-off. Designers should spend more time to make it portable (left side of Fig. 7) or they should spend more time to make it as good as custom design (right side of Fig. 7). For either way, a good script has to have flexible parameterizations (Chang et al., 2018). So, it is not a matter of which tool we would use. Instead, what is more important is whether a designer is willing to use this methodology or not, because analog designers are not generally familiar with such parameterization. In addition, writing a design script requires more skillsets and insights compared to custom designs. As a result, to take full benefit of the reusing, the designers must be patient and be willing to learn something, which is the main hurdle.

Figure 7 Comparison of 3 different frameworks to support reusing process.

Once we overcome the hurdle, there will be more opportunities to further improve the productivity. For example, it allows a machine to accomplish the entire design iteration shown in Fig. 5B (Settaluri et al., 2020). Conventionally, it was believed that the design space is too huge to fully automate the optimization, even with schematic-only simulation. However, recent progress in deep learning technology enable handling such huge space, so a machine can handle the schematic optimization (Hakhamaneshi et al., 2019). However, as mentioned earlier, the layout automation is almost impossible so the machine must struggle with the layout loop. The script-based layout reuse can bridge the gap: (1) The machine sizes the schematic parameters. (2) The layout script generates a layout from the parameters. (3) The machine runs PEX simulations and checks the results. (4) Based on the results, the machine resizes the parameters and repeats (1)–(3) until the circuit is fully optimized. Many efforts should be preceded to fully realize such AI-based design, but it is evident that there will be tons of opportunities along the way.

Memory and storage

Memory scaling limit and 3-D integration

In the previous section, we discussed that the specialization and reuse of the design process will be one of the solutions for the challenges that the logic side is facing. In this section, recent progress and future technology for memory will be presented, and then the opportunities for hardware designers to contribute to the technology innovation will be discussed. In fact, in the memory industry, physics and device engineering have played more critical role compared to design engineering. For example, circuit topology of bit-line sense amplifier in memory module has not been changed for decades while the memory devices have been evolving. This trend is likely to continue in the future, however it is expected that the memory industry will need more innovations from design.

Let us briefly review the challenges that current memory is facing, which is mainly because of the scaling limit as discussed in the “Introduction”. Basically, higher memory density is the top priority which has been enabled by the process scaling. For DRAM, however, lower capacitance due to extensive scaling results in many challenges such as short data retention, poor sensing margin, and interference. As a result, the scaling is no longer as effective as it used to be. Similarly, NAND flash also experiences many non-idealities introduced by the extensive scaling, such as short channel effect, leakage, and interference. Again, the scaling is not useful as it used to be. Recently, however, memory industry has found a very good way rather than pushing the device scaling too hard, they found the solutions from 3D stacking. Figure 8 shows the recent innovations with 3D stacking that have been developed for DRAM and NAND flash, high-bandwidth memory (HBM) and vertical NAND (V-NAND) (Lee et al., 2014; O’Connor, 2014; Tran, 2016; Jun et al., 2017; Xu et al., 2015; Kim, Lee & Kim, 2016; Kim et al., 2009, 2017; Tanaka et al., 2016; Im et al., 2015; Park et al., 2014). In HBM, multiple DRAM dies are stacked, and they are connected by through silicon vias (TSV). A base logic die can be used to buffer between the DRAM stack and the processing unit (host SoC). The logic die and the processing unit are connected through micro-bumps and silicon interposer. Because the memory stack and the processing unit are not integrated in 3D manner, the HBM is often considered as 2.5D integration. Unique features of the HBM such as low capacitance of TSV, 2.5D integration, and high interconnect density of silicon interposer enable high capacity (not always), low power, and high bandwidth compared to legacy DRAM (O’Connor, 2014; Tran, 2016; Jun et al., 2017; Ko et al., 2020). On the other hand, in NAND flash, the memory cells themselves are stacked. Interestingly, nowadays it is higher than 100 layers. In fact, these much of innovations on the capacity, as well as advancements on processing units, burden more on the interconnect side for higher bandwidth and lower latency (Jun et al., 2017; Patterson, 2004; Hsieh et al., 2016). In other word, it requires more contributions from interconnect design so that it is an opportunity for design engineers. For example, as solid-state drive (SSD) capacity has dramatically increased with the V-NAND, the legacy serial-ATA (SATA) interface is not fast enough to provide enough bandwidth. As a result, recent SSD products use NVM Express (NVMe) protocol which is based on peripheral component interconnect express (PCIe) interface. In fact, the PCIe is one of the standards that is evolving very quickly; the industry was working on 16-Gb/s PCIe gen4 in 2016, but started working on 32-Gb/s gen5 since 2018, and 64-Gb/s gen6 specification is going to be released soon (Vučinić et al., 2014; Ajanovic, 2009; Budruk, 2007; Cheng et al., 2010; Li et al., 2018).

Figure 8 Conceptual diagram of (A) HBM and (B) V-NAND.

Since multiple dies are stacked in the HBM, there are more interconnects that are required, and there are unique challenges which can be distinguished from a conventional interconnects, which means there are plenty of works that the interconnect design has to do. For example, the stacked DRAM is communicating with processing unit through the silicon interposer channel, which is quite different to the conventional channels such as channel response and crosstalk (Ko et al., 2020; Liu, Ding & Jiang, 2018). In addition, the stacked DRAM dies are connected by TSV links whose characteristic is also very different (Lee et al., 2015, 2016; Kim et al., 2012). And there is also a logic die where a HBM PHY is used to bridge the DRAM stack and the host SoC. There are also unique issues, for example thermal stability issue due to the stacking (Sohn et al., 2016; Ko et al., 2019), which should be overcome by hardware design.

Introducing new memory devices

In addition to those efforts discuss above, the memory industry is trying to introduce new non-volatile memory (NVM) devices, for example phase-change RAM (PRAM) or resistive RAM (RRAM, also referred to as memristor), whose conceptual diagram is shown in Fig. 9. These devices have only two ports so that it has a smaller footprint of 4F2, and they are able to be integrated in crossbar array and easy to stack (Wong et al., 2010, 2012; Bae et al., 2017; Yoon, Kim & Hwang, 2019; Foong & Hady, 2016; Kau et al., 2009; Yoon et al., 2017; Liu et al., 2013). In addition, they can be formed in back-end process so that they can be integrated on top of the CMOS peripheral circuits, which makes their effective density even higher and realizes a true sense of 3D integration. Moreover, the devices themselves are much faster than NAND device. Note that a faster device means that we need a faster interconnect not to degrade the memory performance due to the interconnect. That is, there will be more demand on high performance interconnect design, similar to what happens on the HBM and V-NAND cases.

Figure 9 New memory device with crossbar array structure.

These devices have many attractive features, however, there are plenty of challenges that need to be overcome to make them succeed in the industry. For example, their operation and side effects are not yet fully modeled; and the PRAM has a reliability issue which is called snapback current during write operation; and the RRAM has a sneak current issue which distorts readout operation as well as write (Yoon et al., 2016); and the variation effect is much larger than the legacy devices because of their intrinsic non-linearity. In fact, these kinds of challenges fall into categories where design engineers can do better than device engineers. For example, they can build a good physics-aware model to bring these devices into an accurate and complex hardware simulation, to enable collaborative optimization between circuits and devices. Because of their non-linearity and hysteresis, some special techniques need to be developed to ensure that they converge in a huge array-level simulation, while capturing the realistic behavior (Bae & Yoon, 2020; Wang, 2017; Kvatinsky et al., 2012; Linn et al., 2014; Chen & Yu, 2015). On the other hand, some circuit design techniques can be introduced to mitigate the snapback current (Kim & Ahn, 2005; Redaelli et al., 2004; Parkinson, 2011). Also, circuit designers can propose variation-tolerant or variation-compensated techniques to address the variation issue (Athmanathan et al., 2016; Park et al., 2017; Hwang et al., 2010; Bae et al., 2018), or sneak-current cancellation scheme for the sneak current issue (Vontobel et al., 2009; Shevgoor et al., 2015; Bae et al., 2016). In addition, looking further forward, RRAM is regarded to be a promising candidate for in-memory computing or neuromorphic computing, because of its capability to store analog weights (Alibart, Zamanidoost & Strukov, 2013; Prezioso et al., 2015; Yoo, 2019; Xue et al., 2019; Kim & Williams, 2019; Yoon, Han & Bae, 2020; Wang et al., 2019). These approaches are believed to overcome the limitation of the current computer architecture, where we need tons of inter-disciplinary research opportunity to realize them.

To summarize this section, the introduction of the 3D integration and the new memory devices is believed to overcome the scaling limit of memory devices, and it needs a lot of supports from hardware designers and gives many opportunities to contribute.

Interconnect

Trend survey and challenges

In this section, the challenges and potential solutions of computer communication interconnect are presented. Recalling the “Introduction”, increase in data and advancement in computing require higher speed interconnect, however the electrical channel becomes more and more inefficient as the data rate increases. Figure 10 shows an architectural diagram of a general interconnect, which serializes parallel input to high-speed non-return-to-zero (NRZ) bitstream and transmits it through electrical channel (wire), and then de-serializes the serial input to parallel at the receive side (Bae, 2020; Chang et al., 2003; Mooney et al., 2006; Bulzacchelli et al., 2006). It is notable that this architecture has not been changed over last 15 years. Since then, the advancements mainly focus on improving building blocks of the given golden architecture, such as designing a better equalizer to provide a better compensation for the channel loss.

Figure 10 Block diagram of general interconnect architecture.

Let us have a deeper look at what causes the challenges on the computer interconnect. As mentioned earlier, the electrical channels do not scale with the silicon technology. However, the interconnect partially takes advantage of the technology scaling, because faster transistors enable a better circuit to overcome the increased channel loss. Figure 11A shows a survey from the state-of-the-art published works (Tamura et al., 2001; Haycock & Mooney, 2001; Tanaka et al., 2002; Lee et al., 2003, 2004; Krishna et al., 2005; Landman et al., 2005; Casper et al., 2006; Palermo, Emami-Neyestanak & Horowitz, 2008; Kim et al., 2008; Lee, Chen & Wang, 2008; Amamiya et al., 2009; Chen et al., 2011; Takemoto et al., 2012; Raghavan et al., 2013; Navid et al., 2014; Zhang et al., 2015; Upadhyaya et al., 2015; Norimatsu et al., 2016; Gopalakrishnan et al., 2016; Shibasaki et al., 2016; Peng et al., 2017; Han et al., 2017; Upadhyaya et al., 2018; Wang et al., 2018; Depaoli et al., 2018; Tang et al., 2018; LaCroix et al., 2019; Pisati et al., 2019; Ali et al., 2019, 2020; Im et al., 2020; Yoo et al., 2020), where we can confirm the correlation between the technology node and the data rate. On the other hand, however, overcoming the increased channel loss has become more and more expensive as the loss is going worse as the bandwidth increases; the equalization circuits consume too much power to compensate the loss, which makes people hesitant to increase the bandwidth. As a result, the tendency has been weakened after 32-nm node. Figure 11B shows the bandwidth trend over years, which evidently shows the bandwidth increase has saturated at around 28–40 Gb/s for years.

Figure 11 Survey and trend of interconnects with respect to (A) technology nodes (B) published years.

Recently, a dramatic change has been made to break the ice. An amplitude modulation technique, which is called 4-level pulse-amplitude modulation (PAM-4), has been adopted in the industry (Upadhyaya et al., 2018; Wang et al., 2018; Depaoli et al., 2018; Tang et al., 2018; LaCroix et al., 2019; Pisati et al., 2019; Ali et al., 2019, 2020; Im et al., 2020; Yoo et al., 2020). With PAM-4, the interconnect can transmit two bits in one-bit period, which doubles the effective bandwidth over NRZ. This dramatic change enables the interconnect bandwidth higher than 50 Gb/s as observed in Fig. 11, and most of latest specifications whose speed are higher than 50 Gb/s employ the PAM-4. In addition, all the golden architecture except for very front-end circuits do not have to be changed with PAM-4, which makes it more attractive.

However, we have to ask if this approach is sustainable or not. We doubled the data rate by adopting PAM-4, then can we do the same with PAM-8 or PAM-16? Fig. 12 shows the comparison between those modulations. The basic concept of PAM-4 is to transmit two bits at the same time, so it achieves 2x higher data rate at the same Nyquist frequency. However, there are 4 signal levels (3 stacked eyes) instead of 2 levels (1 eye), the signal-to-noise ratio (SNR) degrades by 3x, or 9.5 dB. It also introduces some other non-idealities such as non-linearity and CDR complexity, so it can be worse. These days, PAM-4 is justified because the benefit from the higher bandwidth exceeds the SNR loss. We can do the same calculation for PAM-8. It transmits 3 bits while PAM-4 transmits 2 bits, so we get 1.5x higher bandwidth, whereas there are 7 eyes over PAM-4’s 3 eyes, which is equivalent to 7.4-dB SNR degradation. That is, the benefit of PAM-8 is lower than what we can get from PAM-4. The same calculation for PAM-16 is also given in the Fig. 12, where we can find the benefit gets even smaller than PAM-8. From the observation, we can conclude that the amplitude modulation will not be a sustainable solution while the channel capacity and the noise keep the same (Shannon, 1948).

Figure 12 Comparison of (A) NRZ, (B) PAM-4, (C) PAM-8, and (D) PAM-16.

Future directions

As an alternative, we would rather start modifying the golden architecture. One of the potential candidates is a forwarded-clock architecture, which has been explored in several literatures (Casper et al., 2006; Li et al., 2014; Ragab et al., 2011; Casper & O’Mahony, 2009; Hossain & Carusone, 2011; Chung & Kim, 2012; Bae et al., 2016). The bit-error-rate (BER) of an interconnect is a function of the amplitude noise (SNR) and the timing noise (jitter) (Bae et al., 2016). If the SNR becomes worse as the channel loss increases or PAM is used, we can try cancel it out by improving the timing noise. However, in the conventional architecture, way of reducing the timing noise is very limited other than burning more power. Instead, we can forward the transmitter clock to the receiver along with data. Because the timing noise of the forwarded clock and the data are correlated, sampling the data with the forwarded clock cancels the correlated component out hence the effective timing noise at the receive side is minimized. With that, the signaling power and the CDR complexity can be significantly reduced at the same BER, at the cost of just one extra clock channel.

On the other hand, we can also make a bigger change on the architecture. In an analog-to-digital converter (ADC)-based interconnect or digital signal processing (DSP)-based interconnect (LaCroix et al., 2019; Pisati et al., 2019; Ali et al., 2019; 2020; Im et al., 2020; Yoo et al., 2020; Harwood et al., 2007; Chen & Yang, 2010; Wang et al., 2018; Palermo et al., 2018), the analog front-end circuits of the receiver are replaced by a high-speed ADC, and a large fraction of the equalization and CDR stuffs are done in the digital domain. With that, an extensive equalization with dense digital logic is enabled. In addition, PAM-4 justifies the use of ADC because it already requires simple ADC-like front-end as it transmits and receives multiple data levels. The DSP-based interconnect is maturing rapidly these days, however there are still lots of works to come, for example design techniques for building high-speed ADC or resolving high latency of DSP-based receiver.

For a long-term solution, more dramatic change would be required, because the fundamental limit comes from the limited bandwidth of electrical channels. As a result, replacing the electrical channel with optical channel whose bandwidth is almost infinite is believed to be a very promising and eventual solution (Miller, 2000; Young et al., 2009; Jeong, Bae & Jeong, 2017; Thraskias et al., 2018). Conventionally, the optical interconnect has been used for long-distance telecommunication whereas the electrical interconnect is responsible for short-distance computer communication. It is mainly because the optical interconnect consumes much higher power because of power-hungry optical devices and electrical-optical interfaces. On the other hand, because of its lossless nature, the communication distance has little impact on the optical communication performance. However, the electrical interconnect exhibits lower power consumption at short-reach communication, however its power consumption dramatically increases as the communication distance increases because the electrical channel loss increases exponentially with the distance. As a result, there is a critical length where the optical interconnect becomes more efficient than the electrical interconnect, as shown in Fig. 13A (Cho, Kapur & Saraswat, 2004). In similar manner, when the required data rate increases, the power consumption of the electrical interconnects increases exponentially even at the same distance, however it has little impact on the optical interconnect as shown in Fig. 13B. Therefore, the critical length is expected to become shorter as the data rate keep increasing, which make us believe the optical interconnect will be eventually used for computer communication (Cho, Kapur & Saraswat, 2004). However, to realize it, the energy efficiency of optical interconnects must be improved a lot. Currently, the bandwidth-efficiency product of commercial optical interconnects (long-reach) is almost 1,000x lower than that of electrical interconnects (Sun et al., 2020). Then why does a present optical interconnect consume that much power? There can be many reasons, but one of the main reasons is that it is not monolithically integrated. When we look into an optical communication module, there are multiple ICs such as photonics transmitter, receiver, electronic driver IC, retimer IC, and microcontroller. As a result, there are so many interfaces even in a single communication module, where electrical signals come out to the real analog world and experience bulky parasitics, which leads to such poor energy efficiency. That is, monolithic integration where optical devices and VLSI circuits are integrated in a single chip can be a solution for reducing the power consumption (Sun et al., 2015a, 2015b, 2020; Narasimha et al., 2007). In addition to the monolithic integration, dense wavelength division multiplexing (DWDM) enables transmitting multiple data streams through a single optical fiber, which significantly improves the bandwidth density of optical interconnect. DWDM can be regarded as another modulation, but it does not degrade SNR as much as PAM. In fact, it has been more than 30 years ago that the optical interconnect began to gain attention, but there have been no succeed until recently. However, recently, the accumulated efforts are coming out with promising engineering samples such as 5-pJ/bit monolithic DWDM (Sun et al., 2020), 6-pJ/bit 112-Gb/s PAM-4 (Li et al., 2020), so the time will really come soon.

Figure 13 (A) Power comparison between electrical and optical interconnects and definition of critical length. (B) Reduction of critical length at higher speed.

Conclusions

In this paper, the challenges that the current computing system (logic, memory, interconnect) is facing are reviewed. For the logic, the cryptocurrency miners are surveyed which leads to the future direction of specialization, but the downside of specialization is also discussed with an example of a fabless company. For the memory, the challenges and opportunities for design engineering in conjunction with device engineering are reviewed, whereas other reviews tend to focus on devices. For the interconnect, the state-of-the-art works are surveyed, and the recent trends and challenges are discussed. From the reviews and surveys for each part, the solutions and opportunities for those challenges are discussed, which are summarized in Fig. S2. For the logic side, the specialization is proposed for achieving higher efficiency after Moore’s law, and the reusing is also proposed for addressing the productivity issue of the specialization. On the memory side, 3D integration of memory dies or cells and introduction of new NVM devices are expected to overcome the memory density issue. At the same time, they request substantial assistances from design engineers, for example high-performance interconnects, robust physics-aware device modeling, and tons of design techniques to overcome the device limits. Finally, the interconnect side needs to innovate its conventional architecture which has not been changed for a while, and eventually it must drive the optical interconnect.

Supplemental Information

Supplemental Information 1 AI-based AMS circuit design.

Click here for additional data file.

Supplemental Information 2 Summary of future directions to overcome computing challenges.

Click here for additional data file.

Additional Information and Declarations

Competing Interests

Author Contributions

Data Availability

Woorham Bae is employed by Ayar Labs. Woorham Bae is also an Academic Editor for PeerJ.

Woorham Bae conceived and designed the experiments, performed the experiments, analyzed the data, performed the computation work, prepared figures and/or tables, authored or reviewed drafts of the paper, and approved the final draft.

The following information was supplied regarding data availability:

This work does not include any code as it is a literature review.

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
