# Peer review of "Today’s computing challenges: opportunities for computer hardware design"

_PeerJ Computer Science, doi:10.7717/peerj-cs.420_

## Round 0.1 · original submission · Major Revisions

I am sorry for the delay. It has been hard to get reviews back in time due to a variety of reasons. One reviewer was in general happy with your paper but the other reviewer had substantial suggestion to improve the quality of the paper. Please follow the advice and submit the updated paper at your earliest convenience. Thanks for your interest in the journal.

Reviewer 1 ·

Basic reporting

no comment

Experimental design

no comment

Validity of the findings

no comment

Additional comments

Overall, the paper is written well and provides a good survey.

Reviewer 2 ·

Basic reporting

This article really does an interesting job of reviewing three essential aspects of a computing device namely logic, memory and interconnect. Mentioning when the survey was conducted (Sept 2019) is impressive as future readers will get to know the state-of-art till that period. The introduction section provided enough background

It is quite difficult to follow the text in the three sections. It would be very beneficial to readers to put subheadings in each of logic, memory and interconnect sections. A figure/table/chart in the introduction section which summarizes the entire paper would be very beneficial to the readers. An example would be: 
1) Logic
     i)....
     ii)....
2) Memory
     i)....
     ii)....

It is not clear what is the difference between this review article and other ones. It is worth mentioning the key differences with other surveys for each section (logic, memory, interconnect). 

Figures looks convincing and depicts the content mentioned in different sections

Experimental design

The review is conducted in a very narrow space with a few details about each subspace. Few jargons like cryptocurrency syndrome are not quite clear.

Few suggestions to improve the paper: Avoid short (4-5 word) sentences and combine with the immediate ones. This article requires significant changes in terms of writing, clearing grammatical mistakes, and prepositions in sentences like: 1) only simulation, then we draw - incorrect sentence 2) In fact, a better way is reusing, instead of automating. Reuse is a bit easier than automation - two same sentences here are redundant

Validity of the findings

This paper definitely has a positive impact from a reader stand point as it summarized the different aspect of hardware design. The conclusion reflects the introduction and the future directions to explore in each component. It is within the scope of the goals initially set by the authors.

Limited future work and scope is mentioned in the paper which does not give a big picture to the readers to move forward. For example, the paper mentions that the introduction of 3D RAM solves the scaling problem, but does not provide other directions to explore for the researchers to continue in this field.

---

## Round 0.2 · accepted · Accept

Thanks for your interest in PeerJ computer science.

Reviewer 1 ·

Basic reporting

no comment

Experimental design

no comment

Validity of the findings

no comment

Additional comments

no comments

Reviewer 2 ·

Basic reporting

Thanks for adding the subheadings

Experimental design

Thanks for correcting all the sentences.

Validity of the findings

--